# Multi-Aligned and Multi-Scale Augmentation for Occluded Person Re-Identification

**DOI:** 10.3390/s25196210

**Published:** 2025-10-07

**Authors:** Xuan Jiang, Xin Yuan, Xiaolan Yang

**Affiliations:** 1School of Computer Science and Technology, Wuhan University of Science and Technology, Wuhan 430065, China; jiangxuan@wust.edu.cn; 2Hubei Province Key Laboratory of Intelligent Information Processing and Real-Time Industrial System, Wuhan University of Science and Technology, Wuhan 430065, China; 3School of Information and Engineering, Wuchang University of Technology, Wuhan 430223, China

**Keywords:** occluded person re-identification, data augmentation, multi-scale occlusion, crops-mixture

## Abstract

Occluded person re-identification (Re-ID) faces significant challenges, mainly due to the interference of occlusion noise and the scarcity of realistic occluded training data. Although data augmentation is a commonly used solution, the current occlusion augmentation methods suffer from the problem of dual inconsistencies: intra-sample inconsistency is caused by misaligned synthetic occluders (an augmentation operation for simulating real occlusion situations); i.e., randomly pasted occluders ignore spatial prior information and style differences, resulting in unrealistic artifacts that mislead feature learning; inter-sample inconsistency stems from information loss during random cropping (an augmentation operation for simulating occlusion-induced information loss); i.e., single-scale cropping strategies discard discriminative regions, weakening the robustness of the model. To address the aforementioned dual inconsistencies, this study proposes the unified Multi-Aligned and Multi-Scale Augmentation (MA–MSA) framework based on the core principle of ”synthetic data should resemble real-world data”. First, the Frequency–Style–Position Data Augmentation (FSPDA) module is designed: it ensures consistency in three aspects (frequency, style, and position) by constructing an occluder library that conforms to real-world distribution, achieving style alignment via adaptive instance normalization and optimizing the placement of occluders using hierarchical position rules. Second, the Multi-Scale Crop Data Augmentation (MSCDA) strategy is proposed. It eliminates the problem of information loss through multi-scale cropping with non-overlapping ratios and dynamic view fusion. In addition, different from the traditional serial augmentation method, MA–MSA integrates FSPDA and MSCDA in a parallel manner to achieve the collaborative resolution of dual inconsistencies. Extensive experiments on Occluded-Duke and Occluded-REID show that MA–MSA achieves state-leading performance of 73.3% Rank-1 (+1.5%) and 62.9% mAP on Occluded-Duke, and 87.3% Rank-1 (+2.0%) and 82.1% mAP on Occluded-REID, demonstrating superior robustness without auxiliary models.

## 1. Introduction

Person re-identification (Re-ID) aims to match images of the same pedestrian captured from different non-overlapping views, and it is significant for smart transportation management and public security surveillance. However, the existing research [1,2,3,4,5,6,7] mostly relies on idealized assumptions; that is, pedestrians are fully visible in images and their features are clearly distinguishable. In real-world scenarios, pedestrians often suffer from partial occlusion due to vehicles, trees, crowds, etc., which greatly affects the performance of traditional Re-ID methods. For example, when a vehicle occludes the lower body of a pedestrian, the feature extraction will incorporate irrelevant visual elements, such as the vehicle’s contours and colors, contaminating the features and reducing the matching accuracy. In high-density pedestrian scenarios such as subway stations and large gatherings, pedestrians will also occlude each other, confusing appearance features [8,9,10]. Coupled with the dynamic changes in pedestrians’ poses, it becomes even more difficult to distinguish features and establish cross-camera correspondences.

Scarcity of occluded samples in training data is another critical bottleneck for occluded person Re-ID [11]. The sample sizes obtained by the existing dataset construction methods (screening occluded images from existing Re-ID datasets or manually collecting in occlusion scenarios) are limited, restricting models from learning occlusion-invariant representations. The data scarcity prevents models from comprehensively learning feature distribution patterns across different occlusion scenarios during training. When encountering new occlusion types or angles not covered in the training data, the model’s accuracy will drop sharply, and its generalization ability will be compromised.

Previous studies [12,13,14,15] tackle person Re-ID occlusion mainly by predicting pedestrian visibility and extracting features from non-occluded regions. They use auxiliary pre-trained models (e.g., keypoint detectors, semantic parsers, and pose estimators) to isolate visible parts, then apply these features for matching. For example, MAHATMA [12] uses human mask semantics to focus on unoccluded regions, and HOREID [13] leverages keypoint estimation for local features and graph nodes. However, these methods have high computational costs, depend on auxiliary model performance, and cannot solve the core issue of insufficient occluded training samples.

Thus, scholars have explored data augmentation to tackle occlusion, aiming to boost model robustness and generalization, ease sample scarcity, balance performance and efficiency, and avoid auxiliary model limitations [16,17,18]. As shown in Figure 1a, the current occlusion data augmentation methods related to synthetic occlusion can mainly be divided into three categories. The first includes methods like random erasing [19] and OAMN [20]. The former simulates occlusion by randomly selecting a rectangular image area and filling it with random values; the latter does so by cropping background regions from other images and pasting them onto training images. However, the occluded regions lack real object texture, leading models to learn unrealistic features divorced from actual scenes, hindering practical adaptation. The second category, exemplified by SUREID [21], uses real objects (e.g., backpacks and umbrellas) as occluders pasted onto pedestrian images. However, these occluders carry their original backgrounds, introducing extraneous context that is irrelevant to the pedestrian scene. This weakens scenario authenticity and may mislead models to learn occluder backgrounds as discriminative features, diverting focus from key pedestrian features. The third category, represented by DPEFormer [22], improves by using transparent-background occluders to avoid extraneous backgrounds. Yet, flaws remain: the occluder placement is overly simplistic, neglecting real-world spatial logic; more critically, occluders from other datasets often conflict stylistically with person Re-ID data, resulting in mismatched brightness, contrast, or saturation between occluders and pedestrians. This causes models to misinterpret such discrepancies as key cues, failing to learn natural occluder–pedestrian relationships and leading to intra-sample inconsistency, thereby restricting the model’s generalization in real scenarios. Additionally, occlusion causes loss of identity-related information, which cropping-based augmentation can simulate. Random Resized Crop (RRC), from GoogleNet [23] and used in ImageNet [24] classification, generates variants by random cropping to expand datasets. However, in occluded person Re-ID, its randomness risks cutting off key pedestrian details (Figure 1b). Cropped regions may exclude the target entirely, including irrelevant objects or backgrounds, noise that interferes with accurate feature learning and leads to inter-sample inconsistency.

To resolve these dual inconsistencies, we innovatively propose Multi-Aligned and Multi-Scale Augmentation (MA–MSA), a unified framework that integrates the two augmentation methods in parallel, whose core design is illustrated in Figure 2. This framework breaks through the bottlenecks of the existing methods in two regards. First, it proposes the Frequency–Style–Position Data Augmentation (FSPDA) mechanism, consistent with real-world scene distributions. By building an occluder library that includes real-world occlusion distributions, we ensure that the occlusion frequency aligns with the actual conditions of person Re-ID datasets. We achieve style alignment between occluders and the dataset using adaptive instance normalization, and optimize the placement of occluders through hierarchical position rules. This not only guarantees spatial rationality but also enforces space–frequency–style consistency, accurately simulating real occlusion scenarios and fundamentally improving the quality of generated occluded samples. Second, it proposes a two-stage Multi-Scale Crop Data Augmentation (MSCDA) strategy. To tackle the issue of key information loss in single random cropping, we perform multi-scale cropping with non-overlapping ratios in the multi-scale cropping stage, obtaining views of different sizes to deeply explore multi-granular information in images. In the image mixture stage, we adopt linear fusion and regional replacement fusion strategies for dynamic view fusion, which not only avoids information loss but also organically integrates features from different perspectives, further enhancing data diversity and representativeness. The final generated effect diagram of our data augmentation is shown in Figure 1c.

Through the above designs, the MA–MSA framework effectively combines the authenticity of occlusion simulation with the comprehensiveness of crop fusion, significantly improving the quality and diversity of augmented samples and providing higher-quality training data for models to learn robust features in real occluded scenarios. In conclusion, the main contributions of this paper are fourfold:We propose a Frequency–Style–Position Data Augmentation (FSPDA) mechanism to address intra-sample inconsistency: by constructing an occluder library with real-world distributions, achieving style alignment, and optimizing placement positions, it ensures space–frequency–style consistency of occlusion scenarios and improves the authenticity of occluded samples.We design a two-stage Multi-Scale Crop Data Augmentation (MSCDA) strategy to tackle inter-sample inconsistency: it mitigates information loss caused by excessive cropping through multi-scale view fusion and improves data diversity and representativeness.We build a unified MA–MSA framework: Unlike the conventional serial augmentation approach, it integrates FSPDA and MSCDA in parallel. This parallel integration enhances the quality and diversity of augmented samples, providing high-quality training data for models to learn robust features in real occluded scenarios.Experiments on both holistic and occluded datasets show that MA–MSA achieves competitive performance compared to the existing methods, demonstrating superior robustness without auxiliary models.

## 2. Related Works

### 2.1. Occluded Person Re-Identification

**Local Feature-Based Methods.** This category processes human features or feature maps in partitions to train targeted recognition capabilities. The Part-Based Convolutional Baseline (PCB) method [25], as a representative approach for specific predefined semantic parts, divides the human body hierarchy into multiple parts and then trains multiple part-level classifiers. The Consecutive Batch DropBlock Network (CBDB-Net) method [26] evenly divides strips on the feature map and drops each strip one by one to output multiple incomplete feature maps, forcing the model to learn a more robust pedestrian representation in an environment with incomplete information.

**Auxiliary Model-Based Methods.** These methods separate visible body parts from occlusion interference with the help of supplementary information, such as human semantic analysis and pose estimation. The  Mask-Aware Hierarchical Aggregation Transformer (MAHATMA) method [12] guides the model to focus on unoccluded body parts by combining the mask semantic information of the human body. Meanwhile, it proposes a layered HFA module, which aggregates the image patch features processed by Transformer blocks at layers 2, 4, 10, and 12, capturing fine-grained local features by aggregating hierarchical image block representations.

The High-Order ReID (HOReID) method [13] uses a keypoint estimation model to extract semantic local features, regards the local features of images as nodes of the graph, and conveys the relationship information between nodes. The Pose-Masked Feature Branch (PMFB) method [27] obtains the confidence and coordinates of human keypoints through pose estimation, sets thresholds to filter occluded areas, and finally uses visible parts to constrain feature responses at the channel level to solve the occlusion problem. The Pose-Skeleton-Guided Cross-Attention Representation Fusion (PSCR) method [28] obtains the position coordinates, confidence scores, and heatmaps of keypoints through pose estimation, establishes relevant mechanisms to suppress the diffusion of occlusion information, generates fine-grained skeletal region masks to extract comprehensive local features, and fuses different features to enhance their expressiveness and semantic alignment. However, the Adaptive Occlusion-Aware Network (AOANet) method [29] defines four learnable part grabbers and applies cross-attention to motivate them to extract complex semantic information related to body part regions from feature maps. Compared with pose estimation methods, this model does not require additional auxiliary models for pose estimation, resulting in lower complexity.

**Attention Mechanism-Based Methods.** These methods use attention mechanisms to select regions with high significance and differentiation, suppress noise interference, and improve model performance. The  Attention-Aware Compositional Network (AACN) method [30] combines an attitude-guided attention map with a partial visibility score to eliminate background interference and occlusion, and then extracts clean pedestrian features.

The multi-head self-attention (MHSA) method [31] multiplies attention weights with feature maps and applies nonlinear transformations to encourage the multi-head attention mechanism to adaptively capture key local features. However, there is a flaw: the network tends to focus on the most distinctive parts of the image (e.g., a vivid background) while suppressing other areas of the target. The Complete Object Feature Diffusion Network (COFD-Net) method [32] identifies the above attention focus bias problem and makes the features learned by the model truly correspond to the target object by explicitly limiting the distribution of attention.

### 2.2. Occlusion Augmentation

Occlusion augmentation simulates occlusion noise by combining data augmentation methods such as data transformation, thereby improving the model’s sensitivity to occlusion. At present, the proposed occlusion enhancement methods mainly include the following:

**Basic Data Enhancement.** The way to use basic data enhancement is to use erasing, cropping, and flipping to increase the diversity of the data. Random erasure is commonly used to simulate occlusion. By selecting a certain size area in a random position in the image for erasure, it can effectively simulate the situation that pedestrians may be blocked by various objects in reality. RE [19] introduces a random erasure method. Specifically, a rectangular region is randomly selected in the image, and then the pixels in the region are erased with random values, thus generating images with different occlusion degrees, which are used for training the model to extract discriminant identity information of the unoccluded part, thus enhancing the robustness of the model to data changes.

**Cut and Paste.** The Occlusion-Aware Mask Network (OAMN) method [20] proposes a novel occlusion enhancement scheme. In this scheme, the rectangular blocks are clipped randomly from the training image and then scaled to the top, bottom, left, and right positions of the target image to simulate occlusion. In order to simulate more realistic occlusion of misbehavior, the feature erasing and diffusion network (FED) method [33] manually clips blocks of background and occlusion objects from the training image, constructs occlusion sets, and pastes them on the occlusion pedestrian image. The speed-up person ReID (SUREID) [21] pastes small blocks in the occlusion set in the training image. However, this approach introduces additional information about the background of the occlusion map, which may interfere with the training process. The Dynamic Patch-Aware Enrichment Transformer (DPEFormer) [22] involves pasting occluder images obtained from other data sources into the person Re-ID dataset.

However, in the synthetic images generated by this method, the occluders are derived from other datasets, and the intensity distribution of the synthesized occluded images is not uniform enough. In previous works, the design of data enhancement schemes has not been refined enough. Firstly, the simulated noise is often highly random, which causes the generated occlusion samples to deviate significantly from the regular occlusion situations in real-world scenarios, making it difficult to truly replicate the natural forms and distribution characteristics of occlusions in reality. Secondly, during the process of simulating occlusions, additional background noise is likely to be introduced. These non-targeted noises will confuse the model’s learning focus and interfere with its effective capture and learning of real occlusion features. Thirdly, if the occluders used in synthesized images are from other datasets, due to the inherent differences in image style, texture details, etc., between different datasets, it is easy to cause an obvious style gap between the synthesized images and real scenes. At the same time, the intensity distribution of the synthesized occluded images often has uneven problems, which will not only increase the difficulty of model learning but also further restrict the system’s adaptability and generalization performance to various occlusion situations.

## 3. Methodology

One of the challenges of person Re-ID tasks is that there are various occlusions in real-world scenes. Unlike previous occlusion enhancement methods, our method can simulate more realistic occlusion. As shown in Figure 3, our data augmentation scheme consists of two main parts. One is the Frequency–Style–Position Data Augmentation (FSPDA) mechanism based on real-world scene distributions, which is further divided into frequency setting, style setting, and location setting. The other is a two-stage Multi-Scale Crop Data Augmentation Data Augmentation (MSCDA), which [34] includes two main parts: multi-scale cropping and image mixing.

### 3.1. Occlusion Frequency Setting

**Occlusion Library**: To simulate occlusions in the real world, we have established an occlusion set. First, by integrating the distribution patterns and occurrence frequencies of occluders in real-world scenarios, we identified 19 high-frequency common object categories as the core screening criteria through manual statistics and filtering. These categories cover key occluder types, such as pedestrians, various motor vehicles and non-motor vehicles, umbrellas, and backpacks. Based on the aforementioned identified object categories, we performed precise screening and fusion of the base datasets: we extracted samples containing these 19 categories of objects from the training set of the Occluded-Duke [35] dataset and simultaneously selected images covering these object categories from the training set of the COCO [34] dataset. The two types of screened data were integrated to form the base data pool for dataset construction.

Subsequently, the Mask RCNN [36] algorithm was adopted for targeted processing of the object instances in the fused dataset. The core objective was to strip the image background, accurately extract the occluder entities corresponding to the 19 object categories, and synchronously obtain the bounding box information of each occluder. To further improve data quality, manual supplementary correction was additionally conducted on images with suboptimal processing results from the Mask RCNN algorithm (e.g., blurred edges of occluders or incomplete entity extraction), ensuring the extraction accuracy of each occluder entity.

Meanwhile, unlike previous studies that randomly set the proportions of various occluders in the occluder set, we have designed an Occlusion Frequency Module to adjust the occurrence frequency of different types of occluders. Figure 3 illustrates how it adjusts the occlusion set into the occlusion library based on the frequency of occluders.

**Occlusion Frequency Module**: Let there be n = 19 different categories of occluders in the dataset. Suppose the frequency of the i-th category of occluder appearing in the dataset is $f_{i}$ (i = 1, 2, …, m). In our occluder library, we adjust the quantity $N_{i}$ of the i-th category of occluder according to these frequencies $f_{i}$. Specifically, let the total quantity of the occluder library be $N_{total}$. The formula can be expressed as(1)$$\begin{array}{c} N_{i}=N_{total}\times f_{i} \end{array}$$

Figure 4 shows the frequency settings of the highest 8 occluder categories in terms of occurrence frequencies. This method can more realistically reflect the distribution of occluders in real-world application scenarios, thereby improving the generalization ability and practicality of the model. Through this frequency adjustment, our model can better adapt to various occlusion situations, providing a more robust foundation for the subsequent matching steps.

### 3.2. Occlusion Style Setting

Current data augmentation methods predominantly focus on generating diverse samples to boost data diversity. However, these methods overlook the fact that authenticity is also of great importance while ensuring diversity. Authenticity is an indispensable element in data augmentation as it directly impacts the accuracy and applicability of the features learned by the model. If the augmented data significantly deviates from the actual situation, the model may capture some unrealistic or irrelevant features. This limitation is particularly crucial, especially when the objective is to train the model for real-world scenarios. Therefore, we need to carefully consider and validate data augmentation techniques to ensure that the generated data aligns with the natural distribution of real-world images, thereby minimizing the risk of the model learning false patterns that may not generalize well to real-world scenarios.

As shown in Figure 3, our Occlusion Style Transfer Module is as follows: First, input the occlusion image (such as a bench, providing content) and the style image (such as a pedestrian, providing style). An encoder first extracts the content and style features, respectively. Then, ADAIN (adaptive instance normalization) adapts the style features of the style image to the content features of the occlusion image. Finally, a decoder restores and outputs a result image that retains the content of the occlusion and incorporates the target style, achieving content preservation and style transfer.

More specifically, our process is as shown in Algorithm 1. The entire process helps to enhance the authenticity of the augmented data. Since the generated occluder images now conform more closely to the visual style characteristics present in the training set, our model can better learn and adapt to real-world scenarios when dealing with occlusions in the person Re-ID task.
**Algorithm 1** AdaIN-based Image Style Transfer Algorithm**Require:** Occlusion image *x* (providing content information)  1:Style image *y* (providing style information)  2:Pre-trained encoder *e*, Decoder *d***Ensure:** Style-transferred output image *z*  3:**Step 1.** Extract feature maps using the encoder  4:    e(x)← Feature extraction result of occlusion image *x* via encoder *e* (content features)  5:    e(y)← Feature extraction result of style image *y* via encoder *e* (style features)  6:**Step 2.** Perform adaptive instance normalization  7:    Calculate mean and standard deviation of feature maps:  8:        μ(x)← Channel-wise mean of feature map e(x)  9:        σ(x)← Channel-wise standard deviation of feature map e(x)10:        μ(y)← Channel-wise mean of feature map e(y)11:        σ(y)← Channel-wise standard deviation of feature map e(y)12:    Execute AdaIN operation:13:        $z^{\prime }\leftarrow \sigma \left(y\right)\cdot \left(\frac{e(x)-\mu (x)}{\sigma (x)}\right)+\mu \left(y\right)$14:**Step 3.** Generate final image through decoder15:    z← Decoding result of $z^{\prime }$ via decoder *d*16:**return** 
*z*

### 3.3. Occlusion Location Setting

Through a systematic analysis of a large amount of real-scene image data, we found that the locations of common occluders such as cars, bicycles, and pedestrians in the image space exhibit significant distribution patterns. Based on this observation with statistical significance, we designed and constructed the Occlusion Location Determination Module, as shown in Figure 3. For the occluded image library, we divided it into two subsets—the horizontal instance set and the vertical instance set. These two subsets have different preferences for occlusion positions, aiming to more accurately simulate occlusion scenarios in the real world.


**(1) Horizontal Instance Set**


**Bottom Instance Set**: This subset is specifically used to collect occluder instances that frequently appear in the bottom area of the image in real-world scenarios, such as stationary bicycles, street benches, and other typical objects. Let the bottom boundary of the image be represented in the image coordinate system as(2)$$\begin{array}{c} B_{image}=\left\{\left(x,y\right)|y=y_{min},x\in \left[x_{left},x_{right}\right]\right\}, \end{array}$$
where $\left(x_{left},x_{right}\right)$ and $y_{min}$ define the horizontal and vertical boundary ranges of the image, respectively. For each occluder in the subset, the position of its bottom boundary $B_{occluder}$ in the image space is ensured to be precisely aligned with the bottom of the image through the constraint $B_{occluder}\equiv B_{image}$. In the horizontal direction, to simulate the random distribution characteristics of occluders in the actual scene, we introduce a random variable *X* based on a uniform distribution, $X∼U(x_{min},x_{max})$, where $\left[x_{min},x_{max}\right]\subseteq \left[x_{left},x_{right}\right]$. By setting the horizontal position coordinate of the occluder as x=X, on the basis of maintaining the bottom alignment, the random adjustment of the horizontal position is achieved, thereby highly restoring the diverse position states that occluders may appear in at the bottom of the image in the real world.

**Upper Instance Set**: This subset mainly focuses on occluder categories that usually appear in the upper area of the image. Umbrellas are typical representatives. Similarly, we define the top boundary of the image in the image coordinate system as(3)$$\begin{array}{c} T_{image}=\left\{\left(x,y\right)|y=y_{max},x\in \left[x_{left},x_{right}\right]\right\} \end{array}$$

For the occluders in this subset, their top boundary $T_{occluder}$ satisfies the constraint $T_{occluder}\equiv T_{image}$ to ensure precise alignment with the top of the image. In terms of the randomization of the horizontal position, we also introduce a random variable $X^{\prime }$ that follows a uniform distribution, $X^{\prime }∼U\left(x_{min},x_{max}\right)$, and set the horizontal position coordinate of the occluder as $x=X^{\prime }$, where the definition of $\left[x_{min},x_{max}\right]$ is the same as the horizontal range of the bottom instance set. In this way, the actual situations of occluders appearing in different positions in the upper area of the image are effectively simulated.

**Middle Instance Set**: This subset is designed to simulate occluder instances that typically appear in the middle area of the image. Common examples include bags and suitcases. We first define the vertical middle boundary range of the image in the image coordinate system.

Let the vertical middle range of the image be determined by two key lines. The lower boundary of the middle area $y_{\mathrm{mid}\text{-}\mathrm{low}}$ and the upper boundary of the middle area $y_{\mathrm{mid}\text{-}\mathrm{high}}$ divide the image vertically. The middle area in the vertical direction can be defined as $y\in [y_{\mathrm{mid}\text{-}\mathrm{low}},y_{\mathrm{mid}\text{-}\mathrm{high}}]$, where $y_{\mathrm{mid}\text{-}\mathrm{low}}=\frac{y_{\min}+y_{\max}}{3}$ and $y_{\mathrm{mid}\text{-}\mathrm{high}}=\frac{2(y_{\min}+y_{\max})}{3}$ (the specific division ratio can be adjusted according to actual scene characteristics).

The middle boundary constraint of the image for occluders is defined as(4)$$M_{\mathrm{image}}=\left\{\left(x,y\right)\left|y\in \left[y_{\mathrm{mid}\text{-}\mathrm{low}},y_{\mathrm{mid}\text{-}\mathrm{high}}\right],x\in \left[x_{\mathrm{left}},x_{\mathrm{right}}\right]\right.\right\}$$

For each occluder in this subset, its vertical coverage range (e.g., the vertical span of a pedestrian group) must satisfy the constraint that its main body lies within $M_{\mathrm{image}}$. To simulate the random horizontal distribution of occluders in the middle area, similar to the bottom and upper sets, we introduce a random variable $X_{\mathrm{mid}}$ following a uniform distribution,(5)$$X_{\mathrm{mid}}∼U\left(x_{\min\text{-}\mathrm{mid}},x_{\max\text{-}\mathrm{mid}}\right)$$
where $\left[x_{\min\text{-}\mathrm{mid}},x_{\max\text{-}\mathrm{mid}}\right]\subseteq \left[x_{\mathrm{left}},x_{\mathrm{right}}\right]$. By setting the horizontal position coordinate of the occluder as $x=X_{\mathrm{mid}}$, we maintain the vertical middle-area alignment while randomly adjusting the horizontal position. This highly restores the diverse states of occluders that may appear in the middle of the image in real-world scenarios, such as pedestrians randomly distributed horizontally in the middle of a street-view image, helping to comprehensively test and improve the adaptability of image recognition systems to occlusions in different vertical regions.


**(2) Vertical Instance Set**


In this subset, a series of occluder objects appearing in the vertical direction are included, such as tall trees, street billboards, street lamps, etc. Unlike other subsets, the positions of occluders in the horizontal direction of the image are no longer subject to specific restrictions in this subset. However, to ensure the vertical morphological characteristics of the vertical occluder set, it is necessary to process the height of the occluder images. First, based on the total height y of the input image, the height threshold is calculated using the following formula:(6)$$y_{\mathrm{thresh}}=\frac{2}{3}y$$

Then, the occluders are screened or adjusted one by one. If their original height is lower than this threshold, they are processed by means of proportional stretching until their height is not lower than $y_{thresh}$ so as to ensure that these occluders can occupy sufficient vertical space in the image and fully reflect the core characteristics of the vertical morphology.

Meanwhile, a random variable following a uniform distribution $X∼U(x_{\mathrm{left}},x_{\mathrm{right}})$ is introduced to generate horizontal coordinates, and the occluders that meet the height constraint are randomly placed in the left and right regions of the image. This ensures that these occluders can occupy sufficient vertical space in the image, fully reflecting the core characteristics of the vertical morphology and making the occlusion scene closer to the real situation.

### 3.4. Multi-Scale Crop Data Augmentation

In the field of person Re-ID, especially when dealing with occlusion situations, how to effectively augment data to improve the model’s ability to capture features at different scales is a key challenge. The existing single random cropping method has limitations. Due to randomness, it may capture limited or irrelevant information, such as pure backgrounds or unrelated objects. To address this issue, we propose an innovative data augmentation method that combines multi-scale cropping and flexible image mixing strategies.

As shown in Figure 3, our Multi-Scale Crop Data Augmentation follows a two-stage process, including the multi-scale cropping stage and the image mixing stage. Through these two stages, we can fully explore the multi-scale information in the images and effectively fuse it, thereby enhancing the model’s performance in occlusion scenarios.


**(1) Multi-Scale Cropping Stage**


To address the issue that existing single random cropping methods may capture limited or irrelevant information due to randomness, we design multi-scale cropping operations to fully explore multi-scale information in images for multi-scale crop data augmentation.

**Definition of the Cropping Operation Set:** We define a set $O={\left\{o_{m}\right\}}_{m=1}^{M}$ containing *M* cropping operations. The scales of these cropping operations range from an extremely small proportion to covering the entire image, and the scales of each operation do not overlap. Let the size of the original image *I* be H×W (height × width). The cropping ratio corresponding to the *m*-th cropping operation $o_{m}$ is $p_{m}$, so the size of the cropped image is $H_{m}=p_{m}H$ and $W_{m}=p_{m}W$, where $p_{m}$ satisfies $0<p_{1}<p_{2}<\cdots <p_{M}\leq 1$.

To clearly present the “non-overlapping” scale division rule, we set the lower bound of the cropping ratio (minimum cropping ratio) to 0.01 and the upper bound to 1 (corresponding to the scale of the full image). When the number of cropping operations M=3 (this setting is to match the feature hierarchy of “local details–half-body features–global context” in the occluded person re-identification task and achieve hierarchical capture of key information), the total scale range of [0.01,1] is evenly divided into 3 non-overlapping sub-ranges. The scale range of each cropping operation is as follows: (0.01, 0.34) (0.34, 0.67) (0.67, 1).

This division method not only ensures “non-overlapping property” (i.e., the start of each sub-range completely coincides with the end of the previous sub-range; for example, 0.34 is both the end of the 1st cropping and the start of the 2nd cropping), avoiding redundant capture of the same scale information by different cropping operations, but also ensures “effective coverage” (i.e., the 3 sub-ranges fully cover the total scale interval of [0.01, 1] and are distributed in a gradient from low to high scales, enabling comprehensive extraction of local and global features of the image).

**Implementation of Multi-Scale Cropping:** For a given image *I*, we sequentially apply the cropping operations in the set O to obtain a set $V={\left\{r\left(o_{m}\left(I\right)\right)\right\}}_{m=1}^{M}$ containing *M* cropped views. Among them, r(·) is a resizing operation used to adjust all cropped images to a fixed size $H_{\mathrm{fixed}}\times W_{\mathrm{fixed}}$ to meet the input requirements of subsequent models. Small-scale cropping operations can focus on the fine details of the image, which helps to capture the subtle features around the occluded parts of pedestrians in occluded person re-identification; large-scale cropping operations can retain more overall information of the image, which helps to grasp the overall context features of pedestrians.


**(2) Image Mixing Stage**


In this stage, we mix the multiple image views obtained in the multi-scale cropping stage according to specific rules to fuse the image information at different scales. We provide two optional mixing strategies: the linear fusion strategy and the region-replacement fusion strategy.

**Linear Fusion Strategy** For two cropped views $V_{i}$ and $V_{j}$ of the same size, we generate a mixed view $V_{mix-linear}$ through linear interpolation. The formula is as follows:(7)$$\begin{array}{c} V_{mix-linear}=\beta \cdot V_{i}+\left(1-\beta \right)\cdot V_{j} \end{array}$$
where β is the mixing weight, which is randomly sampled from the Beta distribution Beta(γ,γ), and γ is an adjustable hyperparameter. This linear fusion method can smoothly merge the information of the two views, enabling the training samples to contain features from cropped views at different scales and enhancing the generalization ability of the model.**Region-Replacement Fusion Strategy** Given two cropped views $V_{x}$ and $V_{y}$ of the same size, we first randomly generate a binary mask *b* for a rectangular region. Then, we generate a mixed view $V_{mix-replace}$ according to the following formula:(8)$$\begin{array}{c} V_{mix-replace}=b⊙V_{x}+\left(1-b\right)⊙V_{y} \end{array}$$
where ⊙ represents element-wise multiplication. The values of the mask *b* are either 0 or 1, which are used to indicate the positions where the corresponding regions in $V_{y}$ should be replaced into $V_{x}$. Specifically, we first randomly select a rectangular region in $V_{x}$. Let the coordinates of the top-left corner be $\left(x_{start},y_{start}\right)$ and the coordinates of the bottom-right corner be $\left(x_{end},y_{end}\right)$. The area of this region is $A=\left(x_{end}-x_{start}\right)\left(y_{end}-y_{start}\right)$, and it satisfies $A=\theta H_{fixed}W_{fixed}$, where θ is a probability value randomly sampled from the Beta distribution Beta(γ,γ). This region-replacement fusion strategy simulates the changes in local regions of the image, which helps the model to enhance its ability to recognize local features and better handle local occlusion problems in occluded Re-ID.

Through the above multi-scale cropping and flexible image mixing strategies, we construct a new input distribution, which is used as training data. It is expected to significantly improve the performance and robustness of the occluded Re-ID model.

### 3.5. Parallel Integration Mechanism

In existing research, traditional data augmentation methods mostly adopt a serial execution logic: that is, raw samples are processed sequentially according to a preset augmentation pipeline (such as operations like random cropping, brightness adjustment, and rotation) to generate a single augmented sample. Although this serial mode achieves the basic function of data expansion, it has obvious limitations. On the one hand, it can only generate one type of augmented sample at a time, making it difficult to fully cover the diversity of data distribution, which limits the model’s adaptability to different data variants. On the other hand, the temporal dependence of serial operations leads to low overall data processing efficiency. Especially in the scenario of large-scale datasets, it is likely to become a performance bottleneck in the model training process.

Our Parallel Integration Mechanism comprises two independent parallel data augmentation branches: the aforementioned Frequency–Style–Position Data Augmentation (FSPDA) mechanism and Multi-Scale Crop Data Augmentation (MSCDA). These two branches perform simultaneous enhancement on each original input image P, thereby generating two distinct augmented image results in a single processing cycle. This process can be formulated as(9)$$\begin{array}{c} P_{\mathrm{pasted}}=FSPDA\left(P\right),P_{\mathrm{cropped}}=MSCDA\left(P\right) \end{array}$$

Subsequently, these two augmented images, along with the original input image *P*, are fed into a shared-parameter network for feature extraction. Specifically, the shared-parameter network processes *P*, $P_{\mathrm{pasted}}$, and $P_{\mathrm{cropped}}$, respectively, to generate three sets of feature maps. After undergoing the same series of convolutional operations and global pooling, these feature maps are transformed into three independent global feature vectors, denoted as $f_{g}^{1}$, $f_{g}^{2}$, $f_{g}^{3}$. These three global features, each preserving distinct characteristics derived from the original and augmented inputs, are subsequently concatenated or processed as separate entities to participate in the subsequent stages of the network.

### 3.6. Loss Design

We select the classification loss $L_{id}$ and triplet losses $L_{tri}$ for network training. All global and local features within this network are utilized to compute the aforementioned losses. The overall loss function can be formulated as follows:(10)$$L_{final}=\sum_{i=1}^{3}L_{id}\left(p_{g}^{i},y\right)+\sum_{j=1}^{4}L_{id}\left(p_{l}^{j},y\right)+\sum_{i=1}^{3}L_{tri}\left(f_{g}^{i}\right)+\sum_{j=1}^{4}L_{tri}\left(f_{l}^{j}\right)$$
where *p* represents the prediction results of global feature $f_{g}$ and local feature $f_{l}$, *y* represents basic facts.

## 4. Experiment

To comprehensively verify the effectiveness and superiority of the proposed MA–MSA framework in the occluded person re-identification (Occluded Person Re-ID) task, this chapter designs multi-dimensional experiments: First, the adaptability of the model to non-occluded scenarios is verified on mainstream holistic person datasets (Market-1501 and DukeMTMC-reID), and then its occlusion resistance is tested on specialized occluded datasets (Occluded-Duke and Occluded-REID). Meanwhile, ablation experiments are conducted to decompose the contribution of each module, hyperparameter analysis is performed to optimize parameter settings, visualization analysis is carried out to reveal the model’s decision-making mechanism, and performance comparisons are conducted with current mainstream methods. The experiments strictly follow the standard evaluation system in the field of person re-identification to ensure the objectivity and comparability of the results. Ultimately, it is proven that, without auxiliary models, MA–MSA can effectively solve the “dual inconsistency” problem in occluded data augmentation and enhance the robustness and generalization of the model.

### 4.1. Datasets and Evaluation Metrics

**Occluded-REID** [37] represents a specialized image-based dataset for occluded person Re-ID. This dataset was collected using mobile cameras from diverse viewpoints, capturing individuals with various forms of substantial occlusion. Comprising a total of 2000 images, it encompasses 200 distinct people, where each person is represented by 5 full-body images and 5 occluded images. For the experimental setup involving this dataset, the employed framework was trained on the Market-1501 [35] training dataset.

**Occluded-DukeMTMC** [35] is an image-based dataset designed for occluded person Re-ID, which is derived from the DukeMTMC-reID dataset [38]. This dataset provides a comprehensive resource for research, with 15,618 training images featuring 708 individuals. For the testing phase, it offers 2210 query images corresponding to 519 persons and 17,661 gallery images representing 1110 persons. Notably, the dataset has a specific distribution of occluded images: 9% of the training set, all of the query set, and 10% of the gallery set contain occluded images. This unique composition makes it an essential tool for evaluating and developing algorithms capable of handling occluded person Re-ID tasks.

**DukeMTMC-reID** [38] is composed of 36,411 images depicting 1404 individuals, which were taken by 8 cameras. A subset of 16,522 images representing 702 individuals is selected at random as the training set from the dataset, while the rest of the images are distributed into 2228 query images and 17,661 gallery images for testing purposes, ensuring that gallery images do not overlap with the training set.

**Market-1501** [39] stands as an extensively utilized holistic person Re-ID dataset taken by 6 cameras. It comprises 12,936 training images belonging to 751 individuals, along with 3368 query images from 750 individuals, and 19,732 gallery images representing 750 individuals. Non-occluded images predominantly constitute the dataset.

**Evaluation Metrics:** In order to ensure a level playing field for comparing our results with those of existing approaches, we utilize two widely recognized evaluation metrics: Cumulative Matching Characteristic (CMC) curves and mean Average Precision (mAP). These metrics are employed to assess the performance of various Re-ID models. It should be noted that all the experimental evaluations are carried out under the single-query scenario, which helps in maintaining consistency and comparability across different model evaluations.

### 4.2. Implementation Details

We chose a ViT-Base model pre-trained on the ImageNet dataset as the benchmark for our method. The model has 12 transform coding layers, 12 multi-head attention heads, and the eigenvector size is set to 768. Scale each input pedestrian image to 256 × 128. Set the batch size to 32. The model is trained using the Stochastic Gradient Descent (SGD) optimizer. The initial learning rate is set to 0.08, and a cosine learning rate attenuation strategy is applied. Our method is implemented using PyTorch (version 1.8.0), and all experiments are performed on a GeForce RTX 3090 (NVIDIA, Santa Clara, CA, USA). The operating system was Ubuntu 20.04 LTS, with CUDA 11.4 and cuDNN 8.2.4 installed to support GPU acceleration.

### 4.3. Comparison with State-of-the-Art Methods

**Results on Holistic Datasets.** To further validate the efficacy of our proposed method, we conducted comprehensive evaluations on two widely recognized datasets, namely Market-1501 [39] and DukeMTMC-reID [38], benchmarking our approach against several state-of-the-art algorithms. The detailed results are presented in Table 1. On the DukeMTMC-reID dataset, our method delivered outstanding performance, attaining an mAP of 82.9% and a Rank-1 accuracy of 91.2%. Likewise, on the Market-1501 dataset, our method achieved remarkable results, with an mAP of 89.5% and a Rank-1 score of 96.1%.

These findings unequivocally demonstrate that our method exhibits superior performance across all datasets, showcasing formidable competitiveness. Whether confronted with the intricate scenarios inherent in the DukeMTMC-reID dataset or the diverse characteristics of the commonly utilized Market-1501 dataset, our method is capable of accurately performing object identification, thereby validating its efficacy and robustness.

**Results on Occluded Datasets.** In complex real-world application scenarios, an ideal model should possess strong generalization and adaptation capabilities to accurately and efficiently handle various challenges. Based on this, we conducted experiments on the Occluded-Duke [35] dataset and the Occluded-ReID [37] dataset. As shown in Table 2 and Table 3, our method significantly outperforms recent algorithms. On the Occluded Duke dataset, it achieves an mAP of 62.9% and a Rank-1 score of 73.3%; on the Occluded Re-ID dataset, the corresponding mAP is 82.1% and the Rank-1 score is 87.3%, fully demonstrating the high robustness of the model in occluded environments.

This is mainly attributed to the occluded data augmentation module we designed. In person Re-ID, the interference of occlusions and the insufficient proportion of occluded data are prominent issues. This module focuses on improving the quality of augmented images. Compared with traditional solutions, it effectively avoids problems such as poor authenticity of synthesized images, imbalanced intensity distribution, and loss of key information, generating high-quality training data that helps the model to accurately extract pedestrian features. In addition, we innovatively improved the random cropping strategy, replacing single cropping with multi-scale and multiple cropping, which overcomes the drawback of missing key region information and enhances the model’s ability to extract key information at different scales, significantly improving the recognition accuracy and robustness of the model in occluded environments.

### 4.4. Ablation Study

To demonstrate the necessity of each component in the proposed method, we conducted ablation experiments on the Occluded-REID and Occluded-Duke datasets. As shown in rows (b), (c), (d), and (e) of Table 4, models trained with occlusion-related data featuring different operations of location, style, and frequency in the FSPDA component, along with the crop operation in the MSCDA component, achieved better results compared to the baseline (row (a)). When the operations related to location, style, and frequency in the FSPDA component, as well as the crop operation in the MSCDA component, were all enabled (row (f)), the model achieved optimal values in terms of the mAP and Rank-1, fully demonstrating the importance of each component in enhancing the performance of the model for occluded person Re-ID.

### 4.5. Qualitative Analysis

Figure 5 shows the visualization results on the Occluded-REID dataset and Occluded-Duke dataset. Among them, the green boxes represent correct matching results, and the red boxes represent incorrect matching results. For the results in the first two rows of both datasets, it is evident that, compared with the baseline method, the matching results of the method proposed in our paper are more accurate. This is mainly attributed to the more realistic texture mapping method adopted in this paper, which enables the model to simulate more diverse visual scenes closer to the real world, allowing the model to be exposed to a richer variety of samples during training. As a result, the model can adapt to changes in various conditions, such as different lighting, angles, and backgrounds, and can stably identify and match targets when facing complex and changeable situations in practical applications. Meanwhile, the multi-scale cropping technique proposed in this paper exhibits significant advantages. It can adaptively focus on the key areas of the human body. Different from the single or unreasonable cropping strategy used by the baseline method, multi-scale cropping can capture human body information at different scales, effectively avoiding the loss of important information such as body shape and clothing details, thereby extracting more discriminative features to improve matching performance.

In addition, regarding the visualization results in the third row of both datasets, the method proposed in our paper also shows a certain effect in handling person-to-person occlusion. From the perspective of feature interference, when a pedestrian is occluded by others, the features of the occluder’s clothing, accessories, etc., may be similar to those of the occluded person, leading to feature interference. Through analysis at different scales, our multi-scale cropping can better distinguish the features of the occluded person and the occluder: small scales focus on differences in detailed features, while large scales clarify the distinction in overall outlines, thereby reducing the impact of feature interference on recognition. Moreover, in a multi-pedestrian environment, multiple pedestrians are intertwined, with numerous instances of person-to-person occlusion, and the scales and positions of different pedestrians vary. For images processed through multi-scale cropping and fusion, the cropping and fusion of regions at different scales can also effectively simulate multi-pedestrian scenes.

### 4.6. Analysis of the Impact of Hyperparameter γ on Model Performance

In the process of model construction and optimization, we introduce the hyperparameter γ, which plays a crucial role in the data augmentation stage. Specifically, γ is used to define the proportion of images subjected to occlusion processing during data augmentation. To thoroughly explore the impact of γ on model performance, we conducted detailed experiments on our method and the baseline method.

As shown in Figure 6, our method shows a trend of first fluctuating, then rising, and finally slightly declining. When γ is in the range of 0.7–0.8, the Rank-1 accuracy of our method reaches its peak, approaching 87.5. This fully demonstrates that, within this range, our method can achieve the most accurate first-place matching recognition of the target. When the value of γ is small, it means that only a small number of images are subjected to occlusion processing during data augmentation, and our method does not fully exploit and utilize the features, resulting in limited recognition accuracy. When the value of γ is too large, a large number of images are occluded, and the model may introduce too many complex and interfering factors during the learning process, making it difficult to maintain the accuracy at a high level or even causing it to decline.

Focusing on the mAP accuracy index, our method shows a steady upward trend as γ increases. When γ approaches 1, the mAP accuracy reaches approximately 82.2. This strongly indicates that, by reasonably adjusting γ, that is, reasonably controlling the proportion of images subjected to occlusion processing, our method can more efficiently integrate various feature information and significantly improve the comprehensive accuracy of object detection. However, when the value of γ continues to increase beyond a certain extent, the rate of performance improvement gradually slows down, which indicates that the model has approached the optimal state under the current architecture. At this time, further adjusting γ is unlikely to provide substantial performance improvements to the model.

In conclusion, the hyperparameter γ has a profound impact on our method. By skillfully and reasonably regulating γ, that is, scientifically controlling the proportion of images with occlusion processing during data augmentation, our method demonstrates more outstanding optimization potential and superior performance compared to the baseline method in terms of both the Rank-1 accuracy and mAP accuracy indices.

### 4.7. Exploring the Results of Different Mixing Methods on Various Datasets

Table 5 presents the performance of mixup and cutmix techniques on Occluded-REID and Occluded-Duke datasets, measured by mAP and Rank-1 accuracy. On the Occluded-REID dataset, mixup outperforms cutmix, achieving an mAP of 82.1% and a Rank-1 score of 87.3% compared to cutmix’s 81.9% mAP and 86.2% Rank-1. This indicates that mixup is more effective in enhancing feature representation for object Re-ID in this dataset. On the Occlude-Duke dataset, cutmix shows a marginal advantage, with an mAP of 62.9% and a Rank-1 of 73.3%, surpassing mixup’s 62.3% mAP and 72.8% Rank-1. These results highlight the dataset-specific effectiveness of the two mixing methods, suggesting that the choice between mixup and cutmix should be tailored according to the characteristics of the dataset to optimize Re-ID performance.

This performance disparity can potentially be attributed to the differences in the characteristics of the two datasets. The images in the Occluded-REID dataset have relatively clean backgrounds, with minimal interference between the main subjects and the background. In this scenario, the mixup technique, which linearly combines two images and their labels, can preserve the features of the main subjects while smoothing the feature space. This effectively enhances the feature representation ability, thus achieving superior Re-ID performance. Conversely, the Occluded-Duke dataset features more cluttered backgrounds, where complex background information can easily interfere with the extraction of target features. The cropmix technique, by cropping and stitching image regions, can selectively retain and integrate the key features of the target areas. This allows it to effectively avoid the interference caused by the cluttered backgrounds. In such complex backgrounds, cropmix can highlight the target subjects more effectively than mixup, thereby improving the accuracy of person Re-ID.

### 4.8. Experiments of the Values of M and Ratio

To determine the optimal values of *M* and Ratio, we meticulously analyze the experimental results across multiple metrics (mAP and Rank-1) on both the Occluded-REID and Occluded-Duke datasets. As shown in Table 6, the goal is to find the combination that yields the best performance consistently across these datasets as this would indicate the most robust and effective choice for the relevant task.

For M=3 and Ratio=0.01: On the Occluded-REID dataset, an mAP of 82.1 and a Rank-1 of 87.3 are achieved. When M=2 (with any Ratio), the mAP and Rank-1 values are lower (e.g., M=2, Ratio=0.01 has mAP 81.5, Rank-1 84.1). For M=4 (with any Ratio), performance also declines (e.g., M=4, Ratio=0.01 has mAP 81.2, Rank-1 85.7). On the Occluded-Duke dataset, this combination achieves an mAP of 62.9 and a Rank-1 of 73.3. Other *M* values at Ratio=0.01, M=2 (mAP 62.0, Rank-1 73.1) and M=4 (mAP 62.6, Rank-1 73.3) show that M=3 either matches or exceeds in performance.

According to our analysis, this is because, when *M* is 3, the number of croppings is more moderate. It will not lose key information due to too few croppings, nor will it introduce redundancy or interference due to too many croppings. When the Ratio is 0.01, the effective range that can be retained is larger, which helps to make fuller use of the useful content in the image, thereby achieving better performance in person re-identification works.

In conclusion, across both datasets, M=3 and Ratio=0.01 consistently yield the highest or among the highest mAP and Rank-1 values. This superior performance across different evaluation criteria and datasets demonstrates that this combination is the most effective for the experiments conducted, making it the logical choice for further use or analysis in the context of the task.

### 4.9. Complexity Analysis

As shown in Table 7, compared with other models, the proposed MA–MSA model exhibits lower computational complexity, which is mainly reflected in two aspects: (1) The model acquires occlusion-aware capability through occlusion data simulation processing (FSPDA module) and multi-scale crop data augmentation (MSCDA strategy), and, unlike SOTA models, it does not introduce auxiliary models. Existing SOTA methods for occluded person re-identification (such as HOREID [13], PFD [14], etc.) often rely on auxiliary models like human pose estimators and semantic segmentation networks to locate visible regions. In contrast, MA–MSA does not require such external auxiliary models. It only enables the model to fully learn occlusion patterns and feature extraction rules during the training phase through constructing an occlusion sample generation mechanism that conforms to real-world scene distribution (the FSPDA module optimizes occlusion simulation through triple consistency of frequency, style, and position) and a multi-scale view fusion strategy (the MSCDA explores multi-granular information through non-overlapping ratio cropping and dynamic fusion). Without increasing network parameters, MA–MSA achieves the elimination of occlusion noise and the restoration of occluded person features relying on its own module design. Additionally, a comparison of model size and computational complexity between MA–MSA and four recently proposed methods is conducted. Model size and computational complexity are measured by the number of parameters and the FLOP value during the inference process.

### 4.10. Visualization Analysis

We visualize the attention heatmaps via Grad-CAM [52] in Figure 7. As can be seen from the Grad-CAM heatmaps, in the scenarios of the Occluded-Duke and Occluded-REID datasets, compared with the baseline, our method has significant advantages: When pedestrians are occluded by cars, umbrellas, and other objects, the heatmaps of the baseline are often scattered over both the occluders and the human body, while our method makes the heat more focused on the human body region and weakens the interference of the occluders. For example, in the case of pedestrians with umbrellas, the baseline pays more attention to the umbrellas, vehicle bodies, and other occluders, while our method accurately anchors on the human body, extracts effective features, helps the model to more reliably recognize pedestrians in occluded scenarios, and improves the ability to capture key information of the human body in Re-ID tasks. The feature space (t-SNE [53] mapping) of the MA–MSA method on two datasets is shown in Figure 8. It can be clearly seen that MA–MSA makes the positive sample points between pedestrians closer. This proves the effectiveness of our method in solving the occlusion problem.

## 5. Conclusions

To solve “scarce real occluded training samples” and “dual inconsistencies in augmentation” in occluded person Re-ID, this study proposes the MA–MSA framework. The key points are as follows: (1) FSPDA (triple constraints: frequency/style/position) resolves intra-sample inconsistency, boosting synthetic sample authenticity via real occlusion libraries, AdaIN style alignment, and hierarchical placement. (2) MSCDA (multi-scale cropping + dynamic fusion) addresses inter-sample inconsistency, preserving critical features and avoiding single-scale information loss. (3) Parallel FSPDA–MSCDA enables “realistic occlusion + complete feature preservation” optimization. It achieves SOTA performance on occluded datasets (Occluded-Duke: 73.3% Rank-1, 62.9% mAP; Occluded-REID: 87.3% Rank-1, 82.1% mAP) without auxiliary models. Methodologically, the proposed “authenticity–integrity” collaborative augmentation paradigm breaks the limitations of traditional occlusion augmentation, providing a reusable technical framework for high-fidelity data synthesis. Domestically, it advances occluded Re-ID from an “auxiliary model-dependent” paradigm to a “lightweight data-driven” one, reducing technical deployment costs. In practical applications, it enhances the model’s robustness in real occluded scenarios, offering a reliable solution for cross-camera pedestrian matching in intelligent security and smart transportation.

The current framework has limitations in simulating feature changes under complex multi-occluder combinations, leading to reduced recognition accuracy. Future efforts should explore paths to balance “privacy and performance”. Subsequent research can integrate differential privacy and anonymization in dataset construction, expand training data sources via federated learning, and develop methods for adaptive selection of occlusion positions to improve the model’s adaptability to complex occluded scenarios.

## Figures and Tables

**Figure 1 sensors-25-06210-f001:**
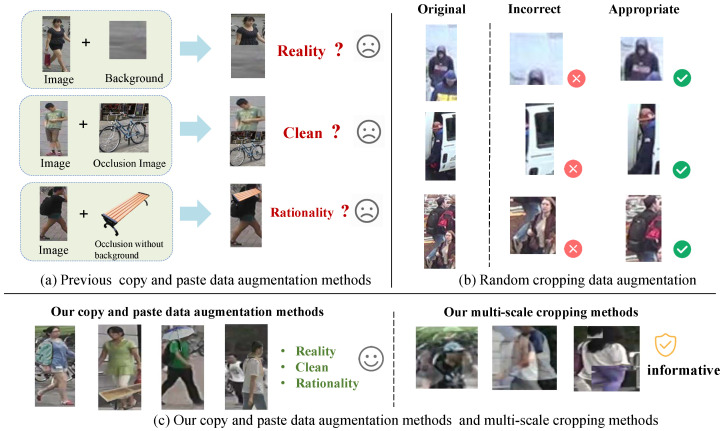
(**a**) Issues in traditional copy–paste augmentation (e.g., ignoring occlusion paste position/style). (**b**) Drawbacks of random cropping augmentation (e.g., generating invalid images). (**c**) Samples of our methods.

**Figure 2 sensors-25-06210-f002:**
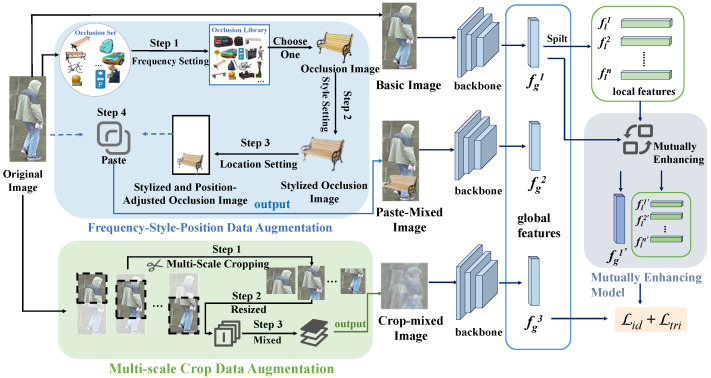
The framework structure diagram of our method. Frequency–Style–Position Data Augmentation (FSPDA) and Multi-Scale Crop Data Augmentation (MSCDA) are the two main parts of our method. We organize the framework in a parallel manner. Among them, FSPDA consists of three components: frequency setting, style setting, and location setting; MSCDA includes two main parts: multi-scale cropping and image mixing.

**Figure 3 sensors-25-06210-f003:**
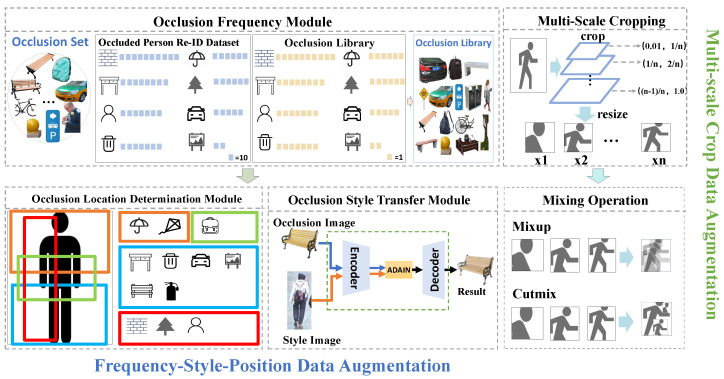
A detailed introduction to our data augmentation method. On the left are the three components and workflow of Frequency–Style–Position Data Augmentation (FSPDA), and on the right are the two main workflows of Multi-Scale Crop Data Augmentation (MSCDA).

**Figure 4 sensors-25-06210-f004:**
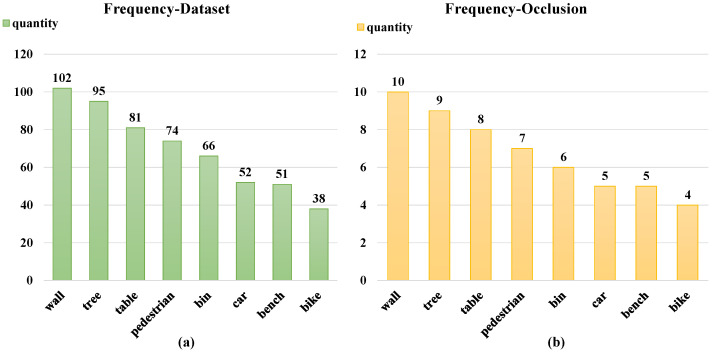
The figure shows the frequency settings of the 8 categories of occluders with the highest occurrence frequencies. Figure (**a**) shows the number of occurrences of each occluder in our statistical dataset, and Figure (**b**) shows the number of occluder settings in our occlusion library.

**Figure 5 sensors-25-06210-f005:**
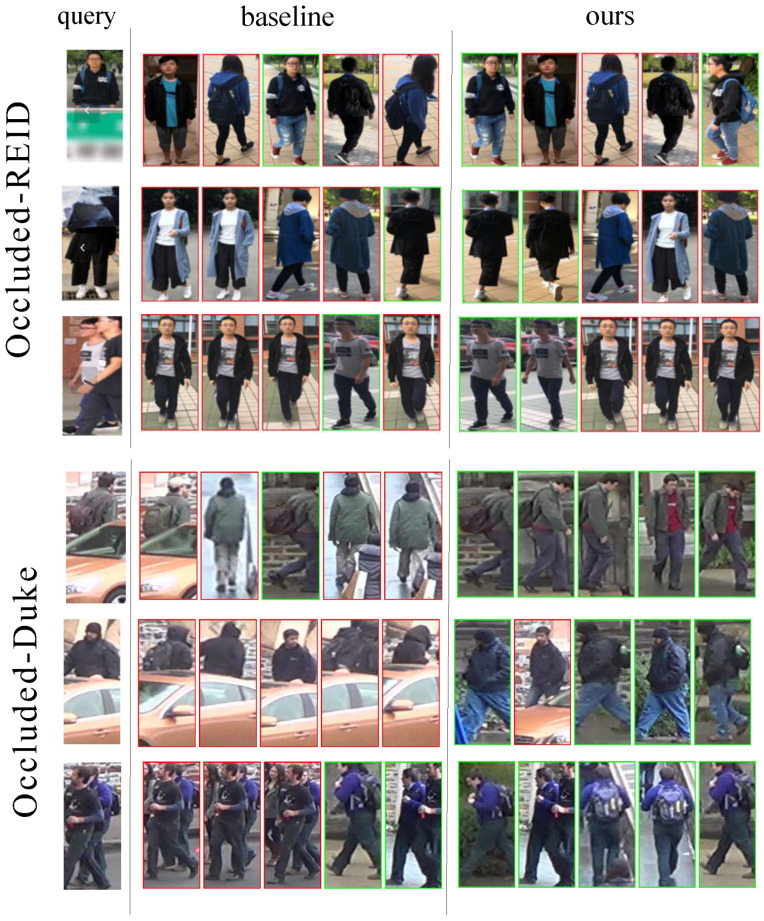
Qualitative analysis of Occluded-REID dataset and Occluded-Duke dataset. The green boxes represent correct matching results, and the red boxes represent incorrect matching results. It is evident that, compared with the baseline method, the results of our proposed method are more accurate.

**Figure 6 sensors-25-06210-f006:**
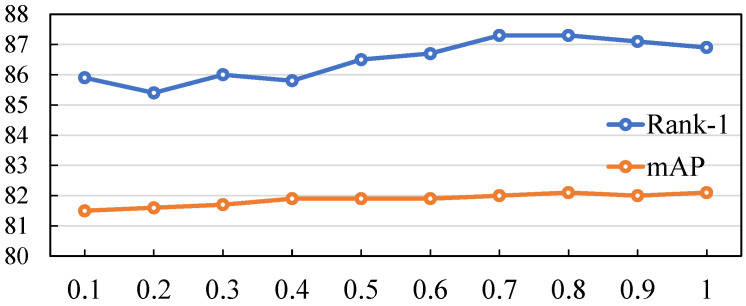
Impact of hyperparameter γ (defining occlusion proportion in data augmentation) on model performance.

**Figure 7 sensors-25-06210-f007:**
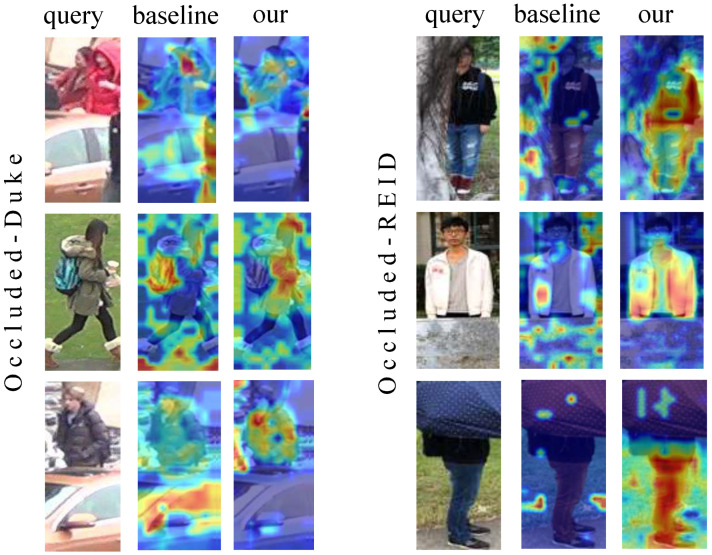
Attention heatmaps of the baseline method and our method by Grad-CAM [52].

**Figure 8 sensors-25-06210-f008:**
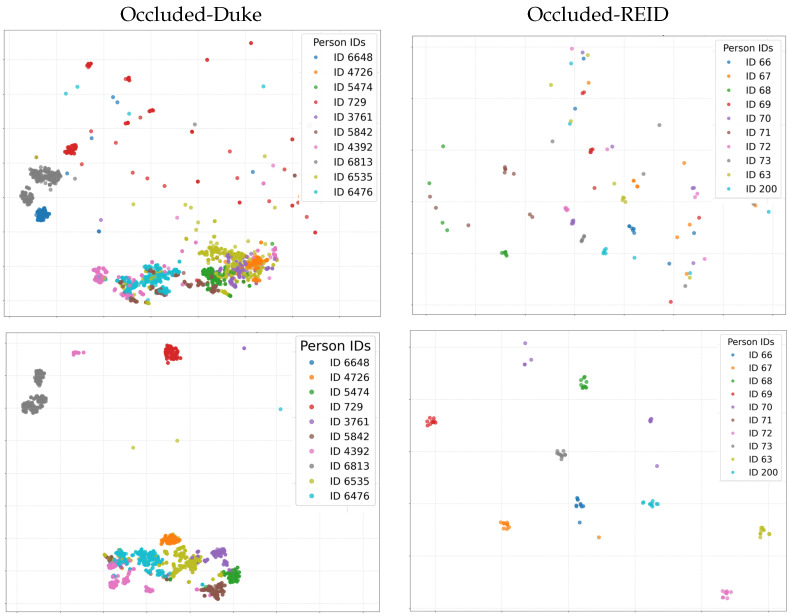
The t-SNE visualization results of 10 identities with the highest occurrence frequencies selected from the Occluded-Duke and Occluded-REID datasets. The first and second rows correspond to the initialization and final well-trained results of our method, respectively.

**Table 1 sensors-25-06210-t001:** Results on holistic datasets. ^†^ means that the results are from our own program runs. Bold indicates the best results.

Methods	Auxiliary Clues	Publication	Backbone	Market-1501		DukeMTMC-reID	
				**Rank-1**	**mAP**	**Rank-1**	**mAP**
PCB [25]	No	ECCV 2018	CNN	92.3	77.4	81.8	66.1
MoS [40]	No	AAAI 2021	CNN	95.4	89	90.6	80.2
TransREID ^†^ [41]	No	ICCV 2021	VIT	95.2	88.9	90.7	82
PAT [42]	No	CVPR 2021	VIT	95.4	88	88.8	78.2
DRL-Net [43]	No	TMM 2022	VIT	94.7	86.9	88.1	76.6
FED [33]	No	CVPR 2022	VIT	95.0	86.3	89.4	78
DPM [44]	No	ACMMM 2022	VIT	95.5	89.7	91.0	82.6
MHSA-Net [31]	No	TNNLS 2022	CNN	94.6	84.0	87.3	73.1
OAT [45]	No	TIP 2024	VIT	95.7	**89.9**	91.2	82.3
PADE ^†^ [46]	No	ICASSP 2024	VIT	95.8	89.7	**91.3**	82.8
PGFA [35]	Yes	ICCV 2019	CNN	91.2	76.8	82.6	65.5
HOREID [13]	Yes	CVPR 2020	CNN	94.2	84.9	86.9	75.6
PFD ^†^ [14]	Yes	AAAI 2022	VIT	95.5	89.7	91.2	83.2
HCGA [47]	Yes	TIP 2022	CNN	95.2	88.4	—	—
bpbreid ^†^ [48]	Yes	WACV 2023	CNN	95.1	87	89.6	78.3
MAHATMA [12]	Yes	TCSVT 2025	VIT	95.2	88.2	91.2	81.2
MA–MSA	No	—	VIT	**96.1**	89.5	91.2	**82.9**

**Table 2 sensors-25-06210-t002:** Results on Occluded-Duke datasets. ^†^ means that the results are from our own program runs. Bold indicates the best results.

Methods	Auxiliary Clues	Publication	Backbone	Rank-1	Rank-5	Rank-10	mAP
PCB [25]	No	ECCV 2018	CNN	42.6	57.1	62.9	33.7
MoS [40]	No	AAAI 2021	CNN	61.0	–	–	49.2
TransREID ^†^ [41]	No	ICCV 2021	VIT	64.2	–	–	55.7
PAT [42]	No	CVPR 2021	VIT	64.5	–	–	53.6
DRL-Net [43]	No	TMM 2022	VIT	65.8	79.3	83.6	53.9
FED [33]	No	CVPR 2022	VIT	68.1	–	–	56.4
DPM [44]	No	ACMMM 2022	VIT	71.4	–	–	61.8
MHSA-Net [31]	No	TNNLS 2022	CNN	59.7	73.2	78.4	44.8
RTGTA [49]	No	TIP 2023	CNN	61.0	69.7	73.6	50.1
OAT [45]	No	TIP 2024	VIT	71.8	–	–	62.2
PADE ^†^ [46]	No	ICASSP 2024	VIT	71.7	83.1	86.4	62.5
PGFA [35]	Yes	ICCV 2019	CNN	51.4	68.6	74.9	37.3
HOREID [13]	Yes	CVPR 2020	CNN	55.1	–	–	43.8
PVPM [50]	Yes	CVPR 2020	CNN	47.0	–	–	37.7
SORN [51]	Yes	TCSVT 2021	CNN	57.6	–	–	46.3
PFD ^†^ [14]	Yes	AAAI 2022	VIT	67.7	80.1	85.0	60.1
HCGA [47]	Yes	TIP 2022	CNN	70.2	83.3	87.0	57.5
bpbreid ^†^ [48]	Yes	WACV 2023	CNN	66.7	81.7	85.5	54.1
MAHATMA [12]	Yes	TCSVT 2025	VIT	73.3	–	–	62.3
MA–MSA	No	—	VIT	**73.3**	**84.4**	**87.1**	**62.9**

**Table 3 sensors-25-06210-t003:** Results on Occluded-REID datasets. ^†^ means that the results are from our own program runs. Bold indicates the best results.

Methods	Auxiliary Clues	Publication	Backbone	Rank-1	mAP
PCB [25]	No	ECCV 2018	CNN	41.3	38.9
MoS [40]	No	AAAI 2021	CNN	—	—
TransREID ^†^ [41]	No	ICCV 2021	VIT	70.2	67.3
PAT [42]	No	CVPR 2021	VIT	81.6	72.1
DRL-Net [43]	No	TMM 2022	VIT	—	—
FED [33]	No	CVPR 2022	VIT	86.3	79.3
DPM [44]	No	ACMMM 2022	VIT	85.5	79.7
MHSA-Net [31]	No	TNNLS 2022	CNN	—	—
RTGTA [49]	No	TIP 2023	CNN	71.8	51.0
OAT [45]	No	TIP 2024	VIT	82.6	78.2
PADE ^†^ [46]	No	ICASSP 2024	VIT	84.7	80.3
PGFA [35]	Yes	ICCV 2019	CNN	80.7	70.3
HOREID [13]	Yes	CVPR 2020	CNN	80.3	70.2
PVPM [50]	Yes	CVPR 2020	CNN	70.4	61.2
SORN [51]	Yes	TCSVT 2021	CNN	—	—
PFD ^†^ [14]	Yes	AAAI 2022	VIT	79.8	81.3
HCGA [47]	Yes	TIP 2022	CNN	88.0	—
bpbreid ^†^ [48]	Yes	WACV 2023	CNN	76.9	68.6
MAHATMA [12]	Yes	TCSVT 2025	VIT	85.8	79.5
MA–MSA	No	—	VIT	**87.3**	**82.1**

**Table 4 sensors-25-06210-t004:** Ablation study results. This table presents the outcomes of ablation experiments on the Occluded-REID and Occluded-Duke datasets. The experiments aim to verify the contribution of each component in the proposed method for occluded person Re-ID. Bold indicates the best results.

Ablation Study	FSPDA			MSCDA	Occluded-REID		Occluded-Duke	
	Location	Style	Frequency	Crop	mAP	Rank-1	mAP	Rank-1
(a)	×	×	×	×	80.3	84.7	62.5	71.7
(b)	✓	×	×	×	81.1	85.5	62.7	72.4
(c)	✓	✓	×	×	81.1	85.7	62.7	72.7
(d)	✓	×	✓	×	81.3	85.9	62.6	72.8
(e)	✓	✓	✓	×	81.5	85.9	62.7	73.0
(f)	✓	✓	✓	✓	**82.1**	**87.3**	**62.9**	**73.3**

**Table 5 sensors-25-06210-t005:** Results of different mixing methods on Occluded-REID and Occluded-Duke datasets.

	Occluded-REID		Occluded-Duke	
	mAP	Rank-1	mAP	Rank-1
mixup	82.1	87.3	62.3	72.8
cutmix	81.9	86.2	62.9	73.3

**Table 6 sensors-25-06210-t006:** Experiments of the values of M and ratio on Occluded-REID and Occluded-Duke datasets. Bold indicates the best results.

M	Ratio	Occluded-REID		Occluded-Duke	
		mAP	Rank-1	mAP	Rank-1
2	0.01	81.5	84.1	62.0	73.1
3	0.01	**82.1**	**87.3**	**62.9**	**73.3**
4	0.01	81.2	85.7	62.6	73.3
2	0.1	78.3	83.6	62.1	73.0
3	0.1	80.8	86.9	62.7	72.9
4	0.1	79.9	84.6	62.5	73.1
2	0.2	77.9	82.5	61.7	72.5
3	0.2	78.5	83.7	61.7	72.8
4	0.2	76.8	81.9	61.6	72.6
2	0.5	77.2	81.8	60.6	71.7
3	0.5	78.1	82.9	60.8	72.0
4	0.5	76.1	80.7	60.5	71.6

**Table 7 sensors-25-06210-t007:** Comparison of different methods in terms of parameters, FLOPs, Rank-1, and mAP.

Methods	Parameters	FLOPs	Rank-1	mAP
PGFA [35]	84.326 M	25.95 G	51.4	37.3
HoReID [13]	144.263 M	22.65 G	55.1	43.8
FED [33]	151.0 M	17.3 G	68.1	56.4
PFD [14]	107.574 M	17.34 G	67.7	60.1
**MA–MSA (Ours)**	**100.127 M**	**16.27 G**	**73.3**	**62.9**

## Data Availability

The Market-1501 dataset can be found using this link: https://github.com/damo-cv/TransReID (accessed on 2 October 2025). The DukeMTMC-reID dataset can be found using this link: https://github.com/damo-cv/TransReID (accessed on 2 October 2025). The Occluded-Duke dataset can be found using this link: https://github.com/lightas/Occluded-DukeMTMC-Dataset (accessed on 2 October 2025). The Occluded-REID dataset can be found using this link: https://github.com/VlSomers/bpbreid (accessed on 2 October 2025).

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
