# Peer review of "Multi-Aligned and Multi-Scale Augmentation for Occluded Person Re-Identification"

_sensors, 2025, doi:10.3390/s25196210_

Round 1
Reviewer 1 Report
Comments and Suggestions for Authors
This paper proposes MA-MSA, a novel data augmentation framework for occluded person Re-ID that integrates FSPDA and MSCDA to address intra- and inter-sample inconsistencies. The method is conceptually clear and technically elegant, generating realistic occlusions while preserving discriminative features, and it achieves state-of-the-art performance on benchmark datasets without auxiliary models. The extensive experiments, strong generalization ability, and practical relevance demonstrate both the substantial workload and notable contribution of this work.
1. While the proposed MA-MSA framework demonstrates strong performance, the paper would benefit from clearer implementation details. Please specify the exact multi-scale cropping ratios used in MSCDA and clarify values such as pₘ (MSCDA) and γ (mixing). A more thorough analysis of other critical hyperparameters—e.g., the number of scales (M) in MSCDA and the AdaIN configuration in FSPDA—would strengthen the evidence of robustness. Additionally, ablation studies for β and b in Equations (10) and (11) are missing and should be included.
2. Several figures mentioned in the text (e.g., Figures 1–3) are not included in the current version. Ensure that all referenced figures are properly inserted, numbered, and cited. Consider adding sample visualizations of augmented images produced by FSPDA and MSCDA to help readers better understand the qualitative effects of your methods.
3. The manuscript states that FSPDA and MSCDA are integrated in parallel, but the exact mechanism remains unclear. Please elaborate on whether both modules process the same batch or different batches, how their outputs are combined, and what selection or fusion logic is applied.
4. The phrase “non-overlapping ratios” in the MSCDA description is ambiguous. Provide explicit definitions or a set of values (e.g., [0.2, 0.4, 0.6, 0.8]) and explain how these ensure non-overlap and effective coverage of the image.
5. Figure 8 appears to have low resolution and a small font size. Please increase both for better readability. Also, review formatting inconsistencies (e.g., “Rank-1” vs. “Rank1”) and misplaced commas. Ensuring consistency will improve the paper’s overall presentation.
6. Minor grammatical errors and awkward phrasings occur throughout the text (e.g., “synthesis must be close to reality” could be revised to “synthetic data should resemble real-world data”). A thorough proofreading by a native English speaker or professional editor is recommended to improve fluency and readability.
Reviewer 2 Report
Comments and Suggestions for Authors
This article tackles one of the fundamental challenges in person re-identification (Re-ID) under occlusion: the disruptive effect of occlusion noise combined with the lack of realistic occluded training data. In real-world surveillance scenarios, individuals are often partially blocked by objects, other people, or environmental structures, making it difficult for algorithms to extract consistent identity features. Existing methods have attempted to address this problem, but they generally fall into two main categories, each with its limitations.
On the one hand, auxiliary model-based approaches depend heavily on pre-trained models such as pose estimators or semantic parsers. While these methods can isolate visible body parts, they significantly increase computational complexity and are highly sensitive to the accuracy of external modules. On the other hand, data augmentation-based approaches attempt to simulate occlusion during training. However, these often generate unrealistic artifacts, stylistic mismatches, and spatial inconsistencies, which ultimately degrade feature discrimination. Such shortcomings lead to two forms of inconsistency: (1) intra-sample inconsistency, caused by misaligned or stylistically mismatched occluders, and (2) inter-sample inconsistency, resulting from random cropping that removes critical identity information.
To address these issues, the authors introduce a unified Multi-Aligned and Multi-Scale Augmentation (MA-MSA) framework, grounded in the principle of realism in data synthesis. Unlike traditional sequential pipelines, MA-MSA integrates two complementary modules in parallel, thereby producing high-quality and diverse training samples:
- Frequency-Style-Position Data Augmentation (FSPDA): This module ensures realism and coherence by aligning frequency components, adapting style with Adaptive Instance Normalization, and enforcing hierarchical rules for occluder placement. It also constructs a realistic occluder library, significantly reducing intra-sample inconsistency.
- Multi-Scale Crop Data Augmentation (MSCDA): To preserve identity features, this module introduces multi-scale cropping with non-overlapping ratios and dynamic view fusion. By retaining multi-granular information, MSCDA mitigates inter-sample inconsistency and reduces the information loss typical of random erasing techniques.
Through the parallel integration of these modules, MA-MSA avoids the drawbacks of sequential augmentation strategies while enhancing both diversity and consistency in training data.
The proposed framework was rigorously evaluated on standard benchmarks. On the Occluded-Duke and Occluded-REID datasets, MA-MSA achieved notable improvements, with Rank-1 accuracies of 73.3% (+1.5%) and 87.3% (+2.0%), respectively, and mAP scores of 62.9% and 82.1%. Importantly, the method also demonstrated competitive results on holistic datasets such as Market-1501 and DukeMTMC-reID, highlighting its robustness beyond occluded scenarios.
Ablation studies further confirmed the necessity of each module, while qualitative visualizations illustrated the framework’s ability to realistically simulate diverse occlusion cases. These findings validate the authors’ claim that realism-based augmentation significantly improves generalization and robustness in Re-ID systems.
When compared with prior work, the contribution of MA-MSA becomes more evident. Traditional auxiliary model-based methods—such as MAHATMA, HOREID, PMFB, and PSCR—show improved feature localization but at the cost of higher complexity and reliance on external accuracy. In contrast, data augmentation-based methods such as Random Erasing, OAMN, SUREID, and DPEFormer introduce occlusion during training but often lack realism, logical placement, or stylistic consistency.
MA-MSA stands out by proposing a parallel integration strategy that enforces frequency, style, and position realism, making it more representative of real-world occlusion. Unlike many augmentation-based methods, it achieves these gains without relying on auxiliary supervision.
The key contributions of this work lie in its unified framework, emphasis on realism in data synthesis, and validated robustness across both occluded and holistic datasets.
Despite its contributions, MA-MSA has several limitations. First, its performance may decline under complex multi-occluder interactions, which are not fully captured by the current occluder library. Second, the framework is primarily focused on static image occlusion, leaving room for future work to extend it toward video-based Re-ID where temporal dynamics and motion occlusions must be considered.
Moreover, the evaluation metrics are mainly restricted to Rank-1 accuracy and mAP. Incorporating additional metrics, such as Rank-5/Rank-10 accuracy, robustness under adversarial noise, and computational efficiency, would provide a more comprehensive performance profile. Finally, while promising, the scalability to large-scale real-world deployment, in terms of computational cost, training efficiency, and model integration into live surveillance systems, remains underexplored.
The MA-MSA framework makes an important step toward realistic and effective augmentation for occluded person re-identification. By integrating FSPDA and MSCDA in a parallel manner, it successfully mitigates the inconsistencies of prior approaches while achieving superior performance across multiple benchmarks. Its emphasis on realism sets it apart from existing literature, though challenges remain in scalability, multi-occluder generalization, and video-based extension. This work thus provides a solid foundation for future research into robust and deployable Re-ID systems.
This work represents an excellent and well-structured contribution to the field of person re-identification under occlusion. It not only introduces a unified framework with strong methodological foundations but also provides thorough experimental validation supported by realistic comparisons with state-of-the-art methods. The paper is clearly written, well-organized, and supported by high-resolution figures that effectively illustrate both the challenges and the proposed solutions. Its comprehensive ablation studies, qualitative analyses, and detailed discussions highlight the robustness and originality of the approach. This is a high-quality and impactful work, and I strongly recommend it for publication.
Reviewer 3 Report
Comments and Suggestions for Authors
This manuscript addresses the problem of occlusion in person re-identification, a challenging and unresolved problem. An augmentation framework for improved occlusion simulation is proposed, which, to the best of my knowledge, is original.
The references are appropriate and up-to-date.
There are some minor grammatical and style issues which could be improved with a final proofreading.
There appears to be a “/” typo in the abstract.
In the abstract, I think some of the wording could be improved for clarity. For example, the mixture of the terms augmentation and synthetic occluders. I would suggest explicitly explaining that synthetic occluders are being used as a method of augmentation here.
Figure 1 isn’t completely clear to me. Parts B (the cross vs tick) and C, in particular. Additional labelling may help.
One of the reasons why re-identification under occlusion is so problematic is that there are very few useful datasets which contain meaningful instances of occlusion. I recommend including a reference in Section 1, paragraph 2 (which starts line 41) to support this:
Topham, L.K., Khan, W., Al-Jumeily, D. and Hussain, A., 2022. Human body pose estimation for gait identification: A comprehensive survey of datasets and models. ACM computing surveys, 55(6), pp.1-42.
Section 2.1 heading typo: “occlude” should be “occluded”
Sections 2.1 and 2.2 are single large paragraphs; I recommend splitting for readability.
In Section 2, I suggest explaining acronyms like “PSCR” and “AOANet”, as well as others, even if they are the names of works in the literature.
The methodology seems sound and is described in great detail. You may wish to consider (optional) moving some details to an appendix or supplementary materials.
Section 2, it would be good to include a summary of the identified gap(s) or limitation(s) which is being addressed in this study.
Figures 2 and 3 should be improved in terms of font size so that all text is readable at 100% view. Similarly, some of the arrowheads are small and easy to miss.
If possible, figures should be closer to where they are referenced. For example, Figure 4.
Line 453, can you provide more details regarding the machine used?
Figure 5 typo “our” -> “ours”
Some acronyms, such as mAP, seem to be defined more than once.
Figure 8 requires improvements in terms of the readability of the text.
Section 3.1 provides more details regarding the occlusion library. What occluders are included? How were they acquired? What processing was implemented?
“n = 19 different categories of occluders” – is this the total number of occluders? Or are there multiple instances within the categories?
The process starting on line 255 should be presented more formally, either as an algorithm or as a figure.
Such methods pose ethical and privacy issues. You may wish to discuss these aspects in your discussion and suggest how they may be addressed (e.g., with de-identification methods).
A good range of relevant datasets has been identified and described. I would recommend including a brief introduction to Section 4.1, explaining the purpose of the datasets. Similarly, you may wish to add a few sentences in the previous section (4.0) to give an overview of the experiments or section (optional).
Could you provide a justification (e.g., supporting citation) for the choice of the benchmark model?
Similarly, how was the initial learning rate of 0.08 chosen?
The conclusion in its current form is a summary of what was done, rather than a conclusion. Try to summarise the main finding (in answer to the research question), and highlight the significance of the findings.
